# An Analysis of Primary Healthcare Antibiotic Prescription Rates Within Castile and Leon (Spain): 2013–2023

**DOI:** 10.3390/antibiotics14111070

**Published:** 2025-10-24

**Authors:** Rocío Salvador Martín, María José Sierra-Medina, María Elena Sánchez-Gutiérrez, Bárbara Rodríguez Vázquez, Noelia Gallego-Ausín, Laura Azofra Casado, Mª Ángeles Machín Morón, Diego Serrano-Gómez

**Affiliations:** 1Grupo de Investigación Ic-SALUD, Facultad de Ciencias de la Salud, Universidad de Burgos, 09001 Burgos, Spain; 2Servicio de Farmacia, Hospital Universitario de Burgos, Sistema Sanitario de Castilla y León, 09006 Burgos, Spain; 3Gerencia de Atención Primaria de Burgos, Sistema Sanitario de Castilla y León, 09006 Burgos, Spain

**Keywords:** anti-bacterial agents, primary health care, Castile and Leon, drug resistance, microbial, rural areas

## Abstract

**Background/Objectives**: According to the World Health Organization, more than one million deaths each year are attributable to bacterial resistance, making it one of today’s main public health threats that could cause up to 10 million deaths by 2050. In this study, antibiotic prescription rates at primary healthcare centers) within Castile and Leon are analyzed over the period 2013–2023, and some of the sociodemographic variables that might influence the prescription of antibiotics are determined. **Methods**: A descriptive, observational, ecological study was conducted based on data gathered by Concylia (pharmaceutical information system—Castile and Leon Health Service). Comparable variables (time of prescription and type of health center) and variables of results (Defined Daily Doses per 1000 health center card-holders per day and qualitative antibiotic selection variables) were analyzed. **Results**: During the first years under analysis, prescription rates increased, followed by a reduction at the start of 2015 that continued up until 2021. Another rise was then recorded, and prescriptions once again reached values in 2023 that were comparable to those observed in 2019 (17.62 and 17.45 Defined Daily Doses per 1000 health center card-holders per day). Throughout Castile and Leon as a whole, there were more prescriptions within urban areas; but when analyzed by provinces, prescriptions were mainly higher in rural areas within most provinces. The percentage of macrolides was higher in urban areas, whereas the percentage of fluoroquinolones was higher in rural areas. **Conclusions:** The variation was repeated throughout the period under study in a similar way in all the provinces of Castile and Leon and at a national level. Different prescription rates were observed by province within the autonomous region. Moreover, higher prescription rates were observed in the rural areas of most provinces. The data source did not allow linking prescriptions to diagnoses, limiting interpretation.

## 1. Introduction

Inappropriate intake of antibiotics is one of the main factors contributing to the development of bacterial resistance. Infections caused by multi-resistant pathogens, one of the major challenges for public health services in the 21st century, are increasingly recognized as a global health emergency [1,2,3,4,5].

Infections due to resistant pathogens are more difficult to treat, leading to lengthier hospital admissions and both higher mortality rates and healthcare costs. The World Health Organization (WHO) has warned that Anti-Microbial Resistance (AMR) is a major global threat. It is estimated that 1.27 million deaths every year may be attributable to AMR [4].

Over the past two decades, global antibiotic consumption has increased, particularly in low- and middle-income countries, while high-income regions have seen modest declines, mainly attributed to stewardship efforts [6]. However, inappropriate use remains widespread, with disproportionate reliance on broad-spectrum antibiotics. In Europe, between 2019 and 2023, the consumption of broad-spectrum and last-resort antibiotics showed an upward trend. The population-weighted mean total consumption rose from 19.8 to 20.0 Defined Daily Doses per 1000 inhabitants per day [7].

Spain has historically been ranked among the European countries with the highest antibiotic prescription rates, despite initiatives for the optimization of anti-microbial treatments, such as the *Plan Nacional frente a la Resistencia a los Antibióticos* (PRAN) [National Plan to combat Antibiotic Resistance]. In 2023, Spain, with 24.1 Defined Daily Doses (DDDs) per 1000 inhabitants per day, had the fourth highest antibiotic prescription rates among all European countries within the European Community [7], so rational use of this sort of treatment assumes a priority focus.

Up until 2023, there had been a global reduction of 14% in human health-related antibiotic prescriptions in Spain since the launch of the PRAN in 2014. The highest prescription rate in 2015 amounted to 28.2 DDDs per 1000 inhabitants per day and was reduced to 24.2 DDDs per 1000 inhabitants per day in 2023 [8]. Nevertheless, significant variations persist between autonomous regions within Spain [9], suggesting that regional and sociodemographic factors may influence the use of antibiotics. A recent study reported a 20.2% increase in the use of anti-microbials between 2017 and 2023 (12.10 versus 14.54 DDDs per 1000 inhabitants per day, respectively). Consumption of antibacterials was higher in women, people over 65 years of age, pensioners, and residents of municipalities with fewer than 10,000 inhabitants [10].

The autonomous community of Castile and Leon is the Spanish region with the largest area and the highest number of provinces, i.e., nine (Figure 1). It is characterized by a wide geographical dispersion, very much higher than in Spain as a whole (regional index 223.9 vs. national index 142.4 in 2024; Table 1) [11], a large proportion of rural areas, and high intra-regional variability.

These geographical attributes could affect the antibiotic prescription profile in primary healthcare (PHC). Previous studies have suggested that prescriptions may be higher in areas with aging populations or in predominantly rural areas [12]. Indeed, evidence has been presented for a degree of intra-regional variability, with prescription rates fluctuating between 13 and 22 DDDs per 1000 inhabitants per day [12].

Around 90% of antibiotic consumption for human health takes place in the context of PHC. Moreover, two-thirds of all patients treated for infectious illnesses receive antibiotic therapy, principally for common respiratory, urinary, and dermatological infections [13,14,15]. According to the available data, between 25% and 30% of the population receives at least one antibiotic treatment every year [16]. Penicillin is one of the most widely used classes of antibacterial drugs for systemic use (especially in combination with beta-lactamase inhibitors), macrolides, and quinolones. In particular, azithromycin (a broad-spectrum macrolide) is now among the most widely dispensed active principles within the region, which makes it a relevant indicator of the likelihood of use [17]. In addition to high rates of prescription and dispensation, excessive or inappropriate administration of these medicines, often in treatments against self-limiting diseases or of viral origin, is causing concern [18,19]. This situation reinforces the need to consolidate and intensify programs that are directed at optimizing the use of anti-microbials in the field of primary healthcare, which is considered a priority area for intervention by the health authorities [13,16,18,19].

While national studies exist, few have analyzed long-term prescription trends at the regional and sociodemographic levels. In this context, this study analyzes PHC antibiotic prescription rates within Castile and Leon from 2013 to 2023, focusing on therapeutic subgroups. By examining temporal and territorial variations, it offers a regional perspective that complements national-level data, highlighting the influence of demographic factors (especially aging) and the impact of stewardship strategies. Despite the impossibility of linking prescriptions with clinical diagnoses, which is inherent in the data source and should be considered when evaluating our results, the findings aim to inform locally adapted interventions and contribute to the broader One Health approach.

## 2. Results

### 2.1. Descriptive Analysis of the Population Holding a Healthcare Card in Castile and Leon

As shown in Figure 2, the percentage of women holding a healthcare card (HCC) was slightly higher than the percentage of men in all the provinces, with the exception of Soria. With regard to the distribution by age, an aging population is evident throughout the community, with slight differences between provinces. The province with the highest number of people over 65 years old was Zamora (31.66%), followed by Leon (28.26%) and Palencia (27.01%). The province with the highest number of people younger than 14 years old was Segovia (11.45%), followed by Burgos (11.11%), Valladolid (10.94%), Avila (10.93%), and Soria (10.85%). The province with the highest number of people between 14 and 65 years old was Segovia (66.46%), followed by Valladolid (65.46%), Soria (64.36%), Burgos (64.17%), and Salamanca (63.93%).

With regard to the rural and urban distribution of the population, the province with the highest percentage of its population registered at a rural PHC center was Segovia (52.64%), followed by Avila (51.94%) and Soria (51.80%), all of which had rural populations over 50%. In contrast, the province with the highest percentage of its population registered at an urban PHC center was Burgos (74.16%), followed by Valladolid (67.23%), Zamora (64.22%), and Salamanca (60.88%), all of which had over 60% of their populations registered at an urban PHC center.

### 2.2. Prescriptions of Anti-Microbials for Systemic Use (J01): Seasonality, Urban–Rural Differences, and Outlier Provinces

There was considerable variation in the prescriptions of anti-microbials for systemic use (J01) at the PHC centers of Castile and Leon according to the quarter of the year. Their prescription rates were higher in the first and last quarters (coinciding with the colder autumnal and winter months) and lower in the third quarter (coinciding with the summer months).

Throughout the years under study, antibiotic prescriptions visibly fluctuated (Table 2, Figure 2). They increased between 2013 and 2015, reaching a peak in the first quarter of 2015 (32.66 and 33.75 DDDs per 1000 HCCs per day in rural and urban areas of Castile and Leon, respectively). Over the period 2016 to 2020, there was a slow descent, with the fewest prescriptions in the second quarter of 2020 (10.80 and 11.20 DDDs per 1000 HCCs per day in rural and urban areas of Castile and Leon, respectively). Subsequently, prescriptions slowly began to rise again until they reached slightly higher figures than in 2019 in the first quarter of 2023 (20.56 and 22.14 DDDs per 1000 HCCs per day in rural and urban areas of Castile and Leon, respectively). The descriptive analysis of antibiotic prescriptions in each province during the period studied (Table 2) shows that the province with the lowest prescription rate was Valladolid every year, while the provinces with the highest prescription rate were Zamora (from 2013 to 2017 and from 2022 to 2023), Soria (from 2018 to 2020), and Ávila (2021). The increase between the provinces with the highest and lowest prescription rates ranged from 32% (Zamora and Valladolid in 2013) to 23.3% (Ávila and Valladolid in 2021).

When analyzing the difference in prescriptions between rural and urban areas (Figure 3), it was observed that prescriptions were very similar in the first years of the study (2013–2015). Subsequently, they were slightly higher at urban PHC centers from 2016, with an average percentile difference over the years under study of 4.35% more prescriptions at urban PHC centers.

The quarterly prescription trends observed in Castile and Leon (Figure 2) were evident in all the provinces. In Figure 4, prescriptions of medication within the J01 subgroup are shown for both the rural and the urban PHC centers of each province of Castile and Leon. Prescriptions were higher in the urban PHC centers of the provinces of Leon, Palencia, and Valladolid, while prescription rates at the rural PHC centers were higher in the other provinces.

When comparing the data on J01 annual prescription rates within the different provinces and throughout all of Castile and Leon with the data on national prescription rates provided by the Spanish Agency of Medicines and Health Products in Spain [17], it was observed that prescriptions at the national level were lower than prescriptions within the region and the provinces (Table 2). The temporal patterns of antibiotic prescriptions for both Spain and Castile and Leon were similar. An increase was recorded between 2014 and 2015, followed by a slow fall over the following years until reaching a minimum with the onset of the pandemic, and then another increase in prescription rates up until 2023. However, the variation at the national level was less striking, as maximums of 17.01 DDDs per 1000 HCCs per day were recorded in 2015 and minimums of 11.61 DDDs per 1000 HCCs per day in 2021. Nevertheless, the maximum was 25.36 DDDs per 1000 HCCs per day in the region of Castile and Leon, and the minimum was 13.04 DDDs per 1000 HCCs per day in 2021. Similar patterns were observed when comparing the data on prescriptions within Castile and Leon recorded by both Concylia and PRAN. The trends were similar, but the variation was lower in those published by PRAN because the maximum values were 19.99 DDDs per 1000 HCCs per day in 2015, and the minimums were 13.36 DDDs per 1000 HCCs per day in 2021.

### 2.3. Differences in Prescription Rates of Pharmacological Therapeutic Subgroups

Table 3 shows a summary of the average prescription rates of medicines over the period 2013–2023 by pharmacological therapeutic subgroup and urban and rural PHC centers of Castile and Leon, as well as their percentile differences.

Prescriptions of medicine in the J01A subgroup were higher at the urban PHC centers in all provinces. Prescriptions of medicine in the J01C subgroup varied to a greater extent and were higher at the rural PHC centers of the provinces of Burgos, Salamanca, Segovia, Soria, and Zamora, whereas the prescriptions were higher at the urban PHC centers of all the other provinces. Prescriptions of medicines in all the other subgroups within the provinces of Avila, Burgos, Segovia, Soria, and Zamora were higher in rural areas (with the exception of J01F in Burgos and J01G in Zamora). In contrast, there were higher prescription rates of all the other subgroups within urban areas of the provinces of Leon, Palencia, and Valladolid (except for J01G in Valladolid). In Salamanca, prescription rates were even more uneven, with higher prescription rates of medicine in the J01E and J01X subgroups within urban areas and higher prescription rates of medicines within rural areas in all the other subgroups.

### 2.4. Prescription Rates of Azithromycin

The prescription rates of azithromycin (J01FA10) in Castile and Leon over the period 2013–2023 (Figure 5, Table 2) were very similar to both the azithromycin subgroup (J01F) (Table 1) and the other group (J01) (Figure 2). An increased number of prescriptions was observed up until 2015, which then fell until 2021 and rose again over the last few years under study to equal the rates in 2018. The increase in the prescription of this antibiotic over the past few years was higher than in earlier cases, resulting in a percentile difference in prescription rates for 2022 with respect to 2021 of 69.05% in rural areas and 62.06% in urban areas).

The prescription rates of azithromycin throughout all of Castile and Leon were greater in urban areas, as it was also in the provinces of Burgos, Leon, Palencia, and Valladolid. Prescription rates were clearly higher in rural areas within all the other provinces (Table 4).

### 2.5. Qualitative Indicators (Choice of Antibiotic)

The analysis of the qualitative indicators revealed different percentile prescription rates for both macrolides and fluoroquinolones.

In the case of macrolides, their prescription rates throughout all of the autonomous region (Figure 6) increased up until 2016, presenting a percentile difference in 2016 with regard to 2015 of around 29%, both in rural and urban areas). From 2016, they presented a slight fall up until 2021, after which the prescription rates increased once again, with a percentile difference between 2022 and 2021 of 21.47% in rural areas and 14.44% in urban areas. In 2023, there was once again a fall in prescription rates.

In the province of Burgos (Figure 7), the prescription rates were unlike those in Castile and Leon as a whole. Within both rural and urban areas, the trends indicate a basic rise throughout the period under study. Despite slight falls in some years, Burgos always maintained higher figures than those of the autonomous region of Castile and Leon, especially since 2019. It is also worth highlighting that the percentages of macrolide prescriptions within rural areas of the province of Leon were higher than the figures for all the other provinces together, up until 2019, when Burgos came to occupy first place.

The comparison between macrolide prescriptions within both rural and urban areas of each province (Figure 7) revealed that Avila and Leon were the only provinces in which prescriptions in rural areas were higher than in urban areas. The percentage prescription rates of macrolides were always higher in urban areas of all the other provinces (Figure 7) and the autonomous region as a whole (Figure 6).

Prescriptions of fluoroquinolones in Castile and Leon (Figure 8) followed a very different pattern from prescriptions of macrolides (Figure 6). A rise was observable up until 2016 that presented a percentile difference with regard to 2015 of approximately 25% in both rural and urban areas of Castile and Leon Subsequently, prescription rates slowly descended over the years up until 2023. The trend was very similar in all the provinces of the autonomous region (Figure 9), with higher figures for the province of Leon (a maximum of 16.09% in 2016 in rural areas) and slightly lower figures for the province of Soria (with a minimum of 6.89% in 2023 in urban areas).

Comparing the percentage prescription rates of fluoroquinolones within rural and urban areas throughout the autonomous region (Figure 8) and in each province (Figure 9), it was observed that fluoroquinolone prescriptions were higher in rural areas throughout all the provinces and all of Castile and Leon.

## 3. Discussion

A descriptive and observational analysis of antibiotic prescription rates at primary healthcare (PHC) centers within the autonomous region of Castile and Leon between 2013 and 2023 forms the core of this study. The prescription rates, therapeutic subgroups, and the active principle azithromycin were analyzed throughout the autonomous region and within each province, comparing both rural and urban areas. Likewise, qualitative indicators (choice of antibiotic), temporal patterns, and rural–urban comparisons between the percentage prescriptions of both macrolides and fluoroquinolones were analyzed.

Overall prescription rates of antibiotics (J01) showed an increase during the first years of this study and began to decline from 2015 onwards across the autonomous region and all its provinces. This reduction may be associated with the launch of the PRAN and its first strategic and action plan (2014–2018) [21], although causality cannot be confirmed without further statistical analysis. The downward trend continued until 2020, when prescription rates dropped sharply, potentially influenced by the COVID-19 pandemic and the related isolation and protective measures implemented during 2020 and 2021 [9,10,15,22]. The fall in prescription rates at a regional level, however, did not happen in the context of hospitals, according to various authors [15,18,23]. From 2021, an increase in prescriptions up to similar figures to those of 2019 was observed [9,15,24].

The differences between the provinces of the autonomous region in relation to the prescription of anti-microbials for systemic use are noteworthy (J01). The provinces of Segovia and Soria, whose populations are concentrated in PHC districts, presented higher prescription rates of DDDs per 1000 HCCs per day. On the contrary, the province of Avila, with the second-highest percentage of its population in rural areas, presented very similar J01 antibiotic prescription rates in both rural and urban areas. In sharp contrast, Burgos, with its population more heavily concentrated in urban areas, presented very similar prescription rates of DDDs per 1000 HCCs per day between rural and urban areas, and in some years, even higher in rural areas. Likewise, prescription rates were higher in rural areas of the provinces of Salamanca and Zamora, whose populations are mainly urban, with a more pronounced difference in the case of Zamora. However, the prescription rates of DDDs per 1000 HCCs per day were higher in urban areas in the provinces of Leon, Palencia, and Valladolid, bearing no relation to the percentage of the population in those areas.

Qualitative variables (choice of antibiotic) [25,26,27] were used to evaluate whether the appropriate antibiotic was prescribed. In this study, two indicators were analyzed, namely, the percentage of macrolide and the percentage of fluoroquinolone prescriptions with regard to total prescriptions. Both indicators are related to broad-spectrum antibiotics, whose use is not part of the first line of attention in PHC [25,26].

The percentage of macrolides (J01FA) included the active principles azithromycin, clarithromycin, erythromycin, spiramycin, and miocamycin [25,26,27,28]. Their use must be limited to treating allergies to Beta-lactam, respiratory infections due to atypical germs, and infection by *Bordetella pertussis*. Due to the high levels of resistance of some microorganisms to this subgroup of antibiotics, such as *Steptococcus pneumoniae*, their use must never be in the first line of PHC [25,26]. With regard to their prescription, different prescription patterns were observed between the provinces. The percentage prescription rates were higher at the urban level in almost all the provinces of Castile and Leon, with the exception of Avila, where urban prescription rates were consistently higher throughout the study. There were likewise sharper differences within Leon in every year, and prescription rates within rural areas varied greatly. Comparing the percentile differences between macrolide prescriptions at the regional level, there was a constant difference of approximately one percentile point throughout the whole period under study, and prescription rates were higher within urban areas.

The fluoroquinolone subgroup (J01MA) groups the following active principles: ciprofloxacin, levofloxacin, moxifloxacin, norfloxacin, and ofloxacin [25,26,27,28]. They can be used to combat respiratory and urinary infections due to their wide spectrum of action. However, they are not part of the first line of PHC treatment due to the high levels of resistance of some microorganisms, including *Escherichia coli* [25,26]. Their use is, in addition, associated with higher risks of infection by *Clostridium difficile* [25,26]. With regard to the prescription of fluoroquinolones, higher figures in rural areas of all provinces of Castile and Leon were observed. At a regional level, the data on prescriptions were 2 percentile points higher at rural health centers throughout the period under study.

The results of this study have, in a general way, shown slightly higher prescription rates of antibiotics for systemic use (J01) at rural healthcare centers within most of the provinces of the autonomous region. Those conclusions are similar to the ones set out by other authors who conducted research throughout Spain [10], Avila [29], Segovia [30], Valladolid [31], and Castile and Leon [12,32]. In the study by Álvarez and others on the consumption of antibiotics within Castile and Leon [12], it was found that the areas with older populations presented higher global prescription rates of antibiotics and of practically all the therapeutic subgroups. An observation that might explain the difference in rates of prescription between the region of Castile and Leon and the data on prescriptions at the national level is that the index of aging in Castile and Leon is very much higher than in Spain as a whole (Table 1) [11]. In many studies, it is highlighted that the major consumers of antibiotics are children and people over 65 years old [12,15,30,31,33,34,35,36,37].

Beyond the sociodemographic differences, other underlying mechanisms may help explain the variability in antibiotic prescription patterns. First, limited access to diagnostic tools in rural areas may lead to more empirical prescriptions, often favoring broad-spectrum antibiotics such as fluoroquinolones [38,39]. Second, the highest prescription rates of macrolides in urban areas could be related to a greater use of emergency services in those areas rather than in rural areas. Urban settings tend to show higher macrolide use, possibly linked to greater reliance on emergency services and time-constrained consultations, which can compromise adherence to clinical guidelines [23,32,40]. That fact has no bearing on worse health in urban zones, but rather on excessive utilization of healthcare services in those zones, which could also contribute to the development of bacterial resistance.

There is therefore great qualitative and quantitative variability in the use of antibiotics at PHC centers within Castile and Leon, with an increase in the prescription of antibiotics for systemic use in rural rather than urban areas within most provinces. That same pattern is repeated in other autonomous regions of Spain [22,41].

Improving the use of antibiotics in the population is a complex task that requires multiple approaches. One of the most important objectives is the education of patients, in order to raise public awareness of such an important problem. So, health campaigns designed to reach the general public are one key point, which, in our opinion, and in the opinions of other authors [12,30,31,33,34,35,36,37], should mainly focus on children and older people as the principal consumers of antibiotics. Other key points to progress in the fight against antibiotic resistance would be the creation of trusting relations between health professionals and patients, proper communication of information between them, and continuous training of health professionals [31,35,36,42].

As well as other health professionals (doctors, pharmacists, …), nurses are fundamental to patient education, as is reflected in the inclusion of the *Organización General de Enfermería* [General Organization of Nursing] in the PRAN [24,43]. Nursing professionals are optimally positioned to guide both individual and group education on the correct use of antibiotics, avoiding their overuse or improper use, and raising awareness of preventive measures against infections, such as washing hands. According to the results covered both in this study and in those of other authors [12,30,31,33,34,35,36,37], this sort of education should be prioritized in rural areas, as well as among children and older people, as those profiles are the highest consumers of antibiotics. This will all contribute to the fight against antibiotic resistance, leading to the achievement of the Agenda for Sustainable Development of the United Nations, with special emphasis on SDG 3 (Health and Well-being), SDG 4 (Quality Education), and SDG 10 (Reduction in Inequality). It will also facilitate the launch of a universal response to action aimed at ending poverty, protecting the planet, and improving the lives and perspectives of people throughout the world.

### Limitations of the Study and Future Lines of Work

This study has some limitations that must be taken into account when analyzing and interpreting its conclusions. In the first place, the data on prescription rates of antibiotics provided by Concylia solely refer to PHC centers within the various provinces of Castile and Leon. The data on prescriptions within hospitals, private healthcare, and any other area outside of the National Health System are excluded, and this may underestimate true antibiotic prescription rates. In addition, the aggregated format of the data provided, without age or center-level detail, prevented inferential statistical analysis, which could limit conclusions, especially about sociodemographic influences. On the other hand, as the data on prescriptions were obtained from electronic prescriptions, it was not possible to determine whether the results are related to the real data on consumption, as correct therapeutic compliance could never be fully validated for such a large sample population.

It must also be taken into account that the veracity of the data provided by Concylia is acceptable, although bias in the processing and grouping of the data may be assumed to exist. It is also worth noting that the provinces are divided into rural, semi-urban, and urban in the Concylia data; however, in this work, each PHC district was grouped into rural and urban, combining semi-urban and urban areas in one single group, which may introduce potential bias.

Another possible limitation is that the data analyzed in this study refer to DDDs per 1000 HCCs per day, based on the format of the data provided by Concylia. On the contrary, the PRAN data on prescription rates at the national level refers to DDDs per 1000 inhabitants per day. It is assumed that the data are equivalent because 100 per cent of the legally resident population in Spain is covered by the Spanish National Health System [44], but potential biases may arise from differences in card-holder registration practices or population mobility. These factors could slightly affect comparability with the national PRAN data and should be considered when interpreting the results.

The source of the information meant that the data on prescription rates were not linked to sociodemographic data; therefore, the data could not link the sex or the age of each patient with the antibiotic prescription. Likewise, no information was found on the disease treated with each antibiotic; therefore, the appropriateness of the prescription could not be determined.

Despite all these limitations, the results obtained in this study present an entry point for related research in the future. The idea of continuing to research the influence of sociodemographic and cultural variables on the consumption of antibiotics appears to be especially interesting. Further research could delve deeper into the study of the causes behind the differences recorded for prescription rates within rural and urban areas, and between the provinces, analyzing which variables might be the most influential. These findings therefore underscore the need for tailored stewardship strategies that consider local healthcare infrastructure and population profiles. On the other hand, the findings also suggest that strategies could be launched to reduce consumption among children and older people. In addition, it might be interesting to expand this study to the area of hospitals and private healthcare for a broader overview of the situation.

## 4. Materials and Methods

A retrospective observational study of antibiotic prescriptions dispensed to the population of healthcare card-holders (HCCs) within all the Basic Health Zones (BHZs) of all the provinces of Castile and Leon was conducted between 2013 and 2023. The data were obtained through the “Concylia Pharmacy Information System. Regional Health Care Management of Castile and Leon” in keeping with the “Protocol on the release of pharmaceutical dispensation data”. The data records the Defined Daily Doses (DDDs), a standard unit of measurement recommended by the WHO for studies on the administration of medicine, which is defined as the assumed average maintenance dose per day for a medicine that is principally indicated for adults and that has a specific route of administration [45].

The WHO Anatomic, Therapeutic, Chemical (ATC) Classification System was used to label the antibiotics [28]. Within this classification, the antibacterials for systemic use are under group J01, which is divided into various subgroups of antibiotics: J01A Tetracyclines; J01B Amphenicols; J01C Beta-lactam Antibacterials, Penicillins; J01D Other Beta-lactam Antibacterials; J01E Sulfonamides and Trimethoprim; J01F Macrolides, Lincosamides, and Streptogramins; J01G Antibacterial Aminoglucosides; J01M Antibacterial Quinolones; J01R Antibiotic Combos; and J01X Other Antibacterials.

The “Antibiotic Prescription Indicators of the Health System of Castile and Leon (Sacyl)” were taken as a reference for dispensation indicators (Sacyl) [25]. Data on the health card posted on the healthcare transparency website by Sacyl [20] were used to determine both the rural and the urban HCC populations.

Antibiotic prescriptions in terms of DDDs per 1000 inhabitants per day were defined in this study as DDDs per 1000 HCCs per day. The same metric has been employed in at least one other study [15]. It is an adaptation of DDDs per 1000 inhabitants per day, as the data on HCCs solely correspond to public health centers, thereby excluding the area of private healthcare, whose users might not hold healthcare cards. In addition, prescriptions of the active principle azithromycin were analyzed because its use was recommended against infection by COVID-19 [46,47].

All the subgroups (J01A, J01C, J01D, J01E, J01F, J01G, J01M, J01R, and J01X) were used for the calculation of the DDDs per 1000 HCC per day of the J01 group, and the following formula was used for azithromycin:
$$DDDs\;per\;1000\;HCC\;per\;day\;=\;\frac{number\;of\;DDDs\;\times \;1000}{number\;of\;HCC\;(1)\;\times \;days\;(2)}$$
(1)The average number of HCCs in each province (by rural and urban) was calculated over 2013–2023.(2)The total was divided by 366 days for the calculations referring to leap years. A total of 91 days were used for the calculations, referring to the first quarter of each leap year.

The data on average prescription rates, together with the standard deviation of each subgroup throughout the period under study, were calculated for each province and for all of Castile and Leon. Trend analysis was only descriptive. The following formula was used to calculate the percentile rural–urban difference:
$$percentile\;rural\;-\;urban\;difference\;=\;\frac{rural\;dispensations-urban\;dispensations}{urban\;dispensations}\times \;100$$

Qualitative variables (choice of antibiotic) are useful to establish the appropriateness of antibiotic prescriptions in relation to the clinical diagnosis and the situation of each patient [25,26]:

-Percentage DDDs of macrolides (J01FA). The use of a macrolide must be linked to very concrete cases. It must not be used as a front-line antibiotic treatment in primary healthcare [25,26]. The following formula was used for its calculation:
$$\%\;DDDs\;of\;macrolides\;=\;\frac{number\;of\;DDDs\;of\;macrolides\;(J01FA)\;\times \;100}{number\;of\;DDDs\;of\;all\;antibiotics\;(J01)}$$


-Percentage DDDs of fluoroquinolones (J01MA). Fluoroquinolones are used to treat respiratory and urinary infections, although they are not included in front-line antibiotic treatment in primary healthcare. In view of the failure of front-line antibiotics, their use must be reduced to very specific cases due to the high levels of antibiotic resistance [25,26]. The following formula was used for its calculation:
$$\%\;DDDs\;of\;fluoroquinolones\;=\;\frac{number\;of\;DDDs\;of\;fluoroquinolones\;(J01MA)\;\times \;100}{number\;of\;DDDs\;of\;all\;antibiotics\;(J01)}$$

As comparative variables, the time of the prescription (in yearly quarters for group J01 and in years for the subgroups) and the type of health center (a binary value: either rural or urban) were analyzed. The instructions in Decree 32/1988 of 18 February, which establishes the territorial boundaries of BHZs within the Regional Community of Castile and Leon [48], subdivides them into rural, semi-urban, and urban. Nevertheless, we grouped semi-urban and urban BHZs into a single group (urban) because residents in both settings have shown similar health behaviors and healthcare access patterns in previous studies in Spain [49] and Castile and Leon [50,51]; and this approach simplifies the interpretation of results.

Only descriptive statistics (mean, standard deviation) were calculated; no inferential analyses were performed.

The Microsoft 365^®^ Excel^®^ program (version 2507; Microsoft Corporation, Redmond, WA, USA) was used for all the calculations.

The Bioethics Committee of the University of Burgos issued a favorable report (IO 14/2004) for this project. No individual-level data were used. All data on antibiotic prescriptions provided through Concylia were anonymized and aggregated and were requested through its “Data Release Protocol”, which contains a “Confidentiality Commitment” and an undertaking to uphold Organic Law 3/2018, of 5 December, on the Protection of Personal Data and the guarantee of digital rights.

## 5. Conclusions

In this study, a comprehensive analysis of antibiotic prescription rates and, therefore, in all likelihood, prescription rates of antibiotics within Castile and Leon was presented over the period 2013–2023. It revealed similarities between the general trend and the records at a national level. An increase was observed in prescription rates up until 2015, with a fall from that year until a minimum was reached in 2020, after which prescription rates once again climbed to similar figures to those of 2019.

It is also noteworthy that the prescription rates of antibiotics for systemic use (J01) and its subgroups in primary healthcare centers within Castile and Leon varied both qualitatively and quantitatively in a remarkable way between the different provinces. And the qualitative indicators under analysis reflected that higher percentages of macrolides were dispensed in urban areas, whereas higher percentages of fluoroquinolones were dispensed. However, prescriptions were higher in rural rather than urban areas in most provinces, even though they were higher in the urban areas of the autonomous region of Castile and Leon as a whole.

The results of this study underscore the need to tailor antibiotic stewardship strategies to demographic and territorial characteristics, prioritizing educational interventions in rural areas and among high-consumption population groups, to improve public health and curb bacterial resistance.

## Figures and Tables

**Figure 1 antibiotics-14-01070-f001:**
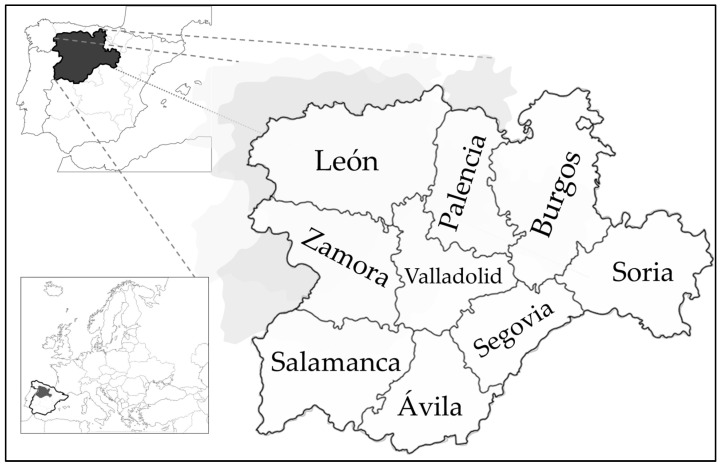
Autonomous community of Castile and Leon. Its location in Spain and Europe, as well as its provincial distribution, are shown.

**Figure 2 antibiotics-14-01070-f002:**
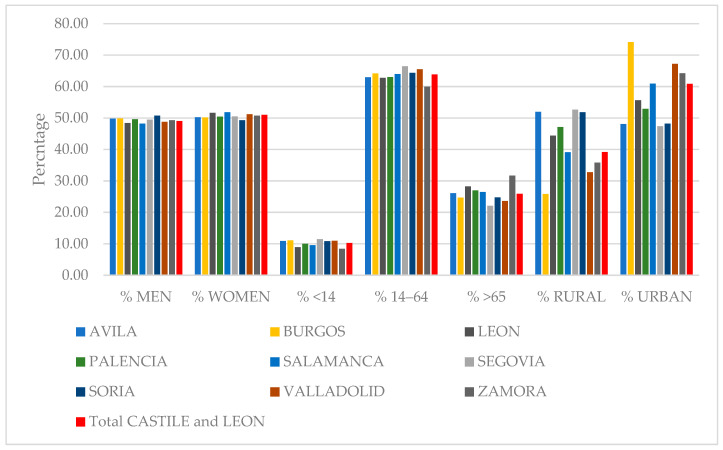
Population distribution by sex, age, and rural or urban BHC of all HCCs within the provinces of Castile and Leon. Authors’ own work based on [20]. BHC: Basic Health Center; HCC: healthcare card-holder.

**Figure 3 antibiotics-14-01070-f003:**
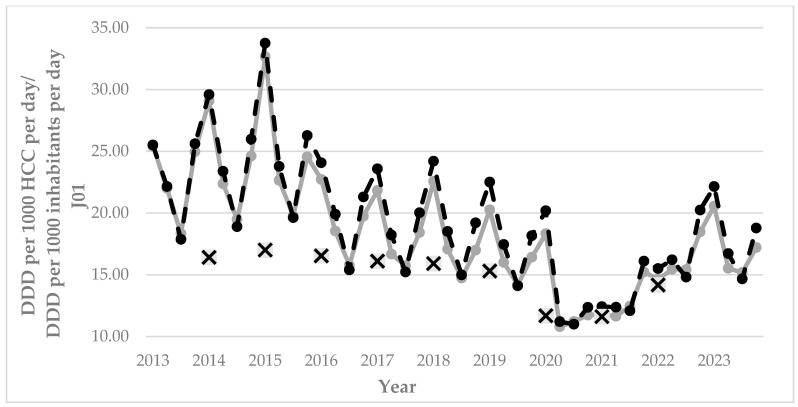
Quarterly prescription rates of anti-microbials for systemic use (J01) (DDDs per 1000 HCCs per day) at both urban (black dashed line) and rural (grey continuous line) PHC centers within Castile and Leon between 2018 and 2023. Annual prescriptions within Spain according to the PRAN, presented as DDDs per 1000 inhabitants per day, are shown as a reference benchmark (squares with crosses), for which data were only available for 2014 to 2022. DDDs: Defined Daily Doses; HCCs: healthcare card-holders. PRAN: Plan Nacional frente a la Resistencia a los Antibióticos [National Plan to combat Antibiotic Resistance].

**Figure 4 antibiotics-14-01070-f004:**
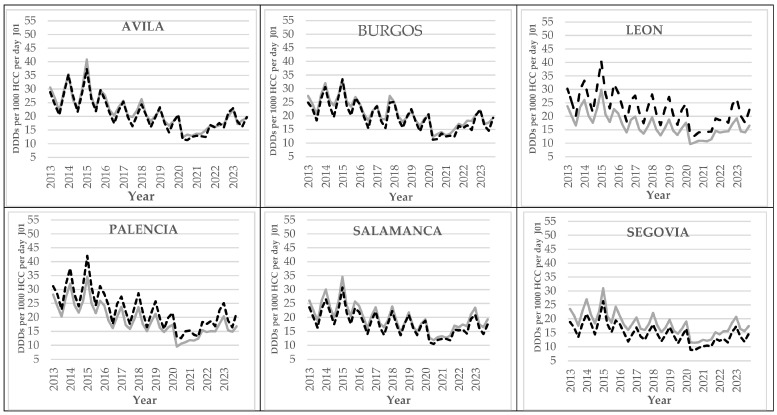
Graphs showing quarterly J01 antibiotic prescription rates (anti-microbials for systemic use) at both urban (black dashed line) and rural (grey continuous line) PHC centers within the province of Castile and Leon (DDDs per 1000 HCCs per day) between 2013 and 2023. DDDs: Daily Defined Doses; HCC: healthcare card-holder.

**Figure 5 antibiotics-14-01070-f005:**
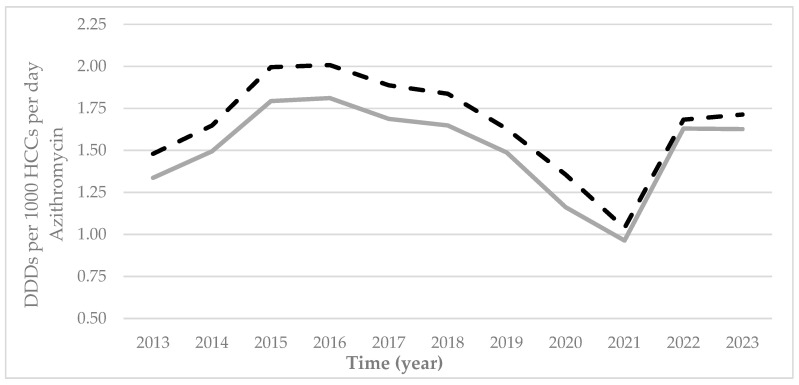
Prescription rates of azithromycin through the PHC centers of Castile and Leon (DDDs per 1000 HCCs per day) by urban (black dashed line) and rural (grey continuous line) areas between 2013 and 2023. DDDS: Defined Daily Doses; HCCs: healthcare card-holders.

**Figure 6 antibiotics-14-01070-f006:**
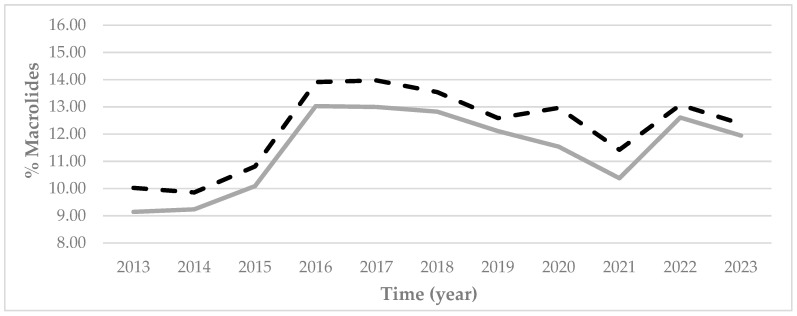
Percentage prescription rates of macrolides at the PHC centers within Castile and Leon by urban (black dashed line) and rural (grey continuous line) areas between 2013 and 2023.

**Figure 7 antibiotics-14-01070-f007:**
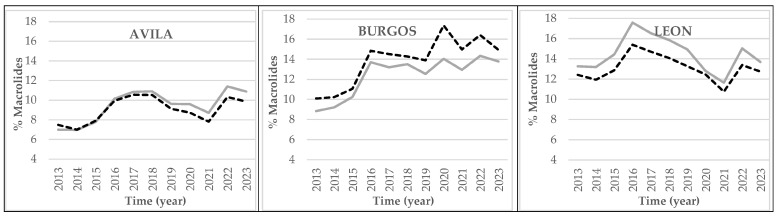
Percentage prescription rates of macrolides (DDDs per 1000 HCCs per day) at the PHC centers of each province of Castile and Leon by urban (black dashed line) and rural (grey continuous line) areas between 2013 and 2023.

**Figure 8 antibiotics-14-01070-f008:**
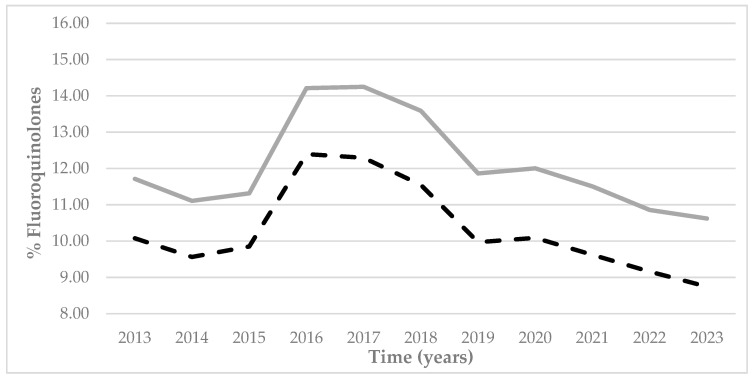
Percentage prescription rates of fluoroquinolones) at the PHC centers within Castile and Leon by urban (black dashed line) and rural (grey continuous line) areas between 2013 and 2023.

**Figure 9 antibiotics-14-01070-f009:**
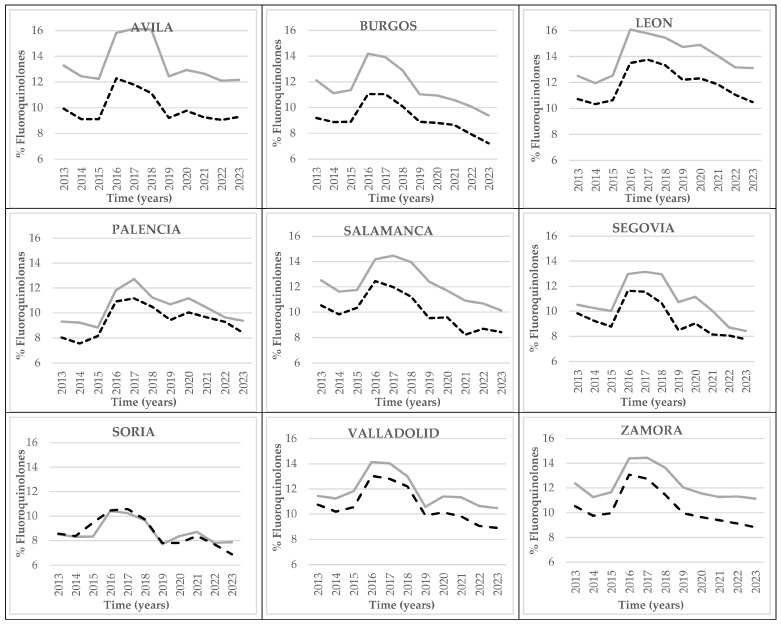
Comparisons of percentage prescription rates of fluoroquinolones (DDDs per 1000 HCCs per day) at PHC centers within each province of Castile and Leon, by urban (black dashed line) and rural (grey continuous line) areas between 2013 and 2023.

**Table 1 antibiotics-14-01070-t001:** Comparison of the aging index ^1^ in Spain and in Castile and Leon over the period 2013 to 2024 [11].

	2013	2014	2015	2016	2017	2018	2019	2020	2021	2022	2023	2024
Spain	109.9	112.6	114.7	116.3	118.4	120.6	123	125.8	129.2	133.6	137.3	142.4
Castile and Leon	179.4	182.9	185.4	188	191.2	194.5	198.3	202.3	205.7	212.9	217.2	223.9

^1.^ Aging index: percentage of the population over 65 years old in relation to the population under 16 years old on 1 January of any one year.

**Table 2 antibiotics-14-01070-t002:** Comparison of annual prescriptions of J01 (anti-microbials for systemic use) at PHC centers within each province in Castile and Leon (DDDs per 1000 HCCs per day) and in the autonomous region, and the data published by PRAN for Castile and Leon and Spain (DDDs per 1000 inhabitants per day). AV: Avila; BU: Burgos; LE: Leon; PA: Palencia; SA: Salamanca; SE: Segovia; SO: Soria; VA: Valladolid; ZA: Zamora; C&L: Castile and Leon; ES: Spain; PRAN: Plan Nacional frente a la Resistencia a los Antibióticos (National Plan for combating Antibiotic Resistance); n.a.: not available.

Year	AV	BU	LE	PA	SA	SE	SO	VA	ZA	C&L	C&L PRAN	ES
2013	26.28	23.43	23.91	26.73	21.68	19.09	24.87	18.55	27.31	22.66		
2014	28.24	25.47	25.12	28.16	23.11	20.50	25.60	19.80	29.00	24.13	18.94	16.42
2015	29.31	26.13	27.18	29.43	24.38	21.72	26.49	20.80	30.23	25.36	19.99	17.01
2016	22.53	20.39	21.14	21.85	19.00	16.58	21.10	16.57	23.16	19.71	19.86	16.54
2017	21.16	19.60	19.60	20.80	18.17	16.22	20.62	15.89	21.86	18.75	19.18	16.10
2018	20.63	19.90	19.21	20.44	18.12	16.35	21.62	15.75	20.80	18.59	19.11	15.91
2019	18.53	18.30	18.88	18.75	17.28	15.36	20.05	14.54	19.58	17.45	18.08	15.31
2020	14.27	14.32	14.27	13.87	13.47	12.18	15.31	11.13	14.40	13.36	13.80	11.68
2021	14.18	13.71	13.86	14.02	13.45	12.03	13.89	10.88	13.82	13.04	13.36	11.61
2022	17.60	17.11	17.69	17.34	16.84	14.55	18.08	13.61	17.84	16.39	16.70	14.17
2023	19.24	18.24	19.07	18.70	17.90	15.94	19.13	14.71	19.21	17.62	n.a.	15.26

**Table 3 antibiotics-14-01070-t003:** Average prescription rates of medicine by therapeutic subgroup at the PHC centers of Castile and Leon, by province and by rural and urban areas over the period of study (2013–2023). % DIF R-U: percentile rural–urban difference.

		J01A	J01C	J01D	J01E	J01F	J01G	J01M	J01X
AVILA	R	0.71 (±0.07)	12.60 (±4.10)	2.07 (±0.27)	0.61 (±0.07)	2.05 (±0.38)	0.017 (±0.003)	2.91 (±0.78)	0.49 (±0.05)
	U	0.95 (±0.09)	12.80 (±4.10)	1.86 (±0.31)	0.51 (±0.04)	1.91 (±0.40)	0.014 (±0.008)	2.06 (±0.53)	0.48 (±0.02)
% DIF R-U		−25.76	−1.57	11.10	20.26	7.17	28.13	41.03	3.42
BURGOS	R	0.53(±0.05)	11.84 (±3.80)	2.24 (±0.31)	0.40 (±0.09)	2.55 (±0.32)	0.004 (±0.003)	2.41 (±0.63)	0.59 (±0.06)
	U	0.73(±0.05)	11.21 (±3.60)	2.08 (±0.24)	0.36 (±0.06)	2.67 (±0.28)	0.002 (±0.001)	1.78 (±0.45)	0.53 (±0.03)
% DIF R-U		−27.24	5.57	7.52	11.69	−4.56	82.15	35.00	12.81
LEON	R	0.58 (±0.02)	8.72 (±2.84)	1.80 (±0.32)	0.46 (±0.04)	2.50 (±0.58)	0.004 (±0.003)	2.34 (±0.41)	0.45 (±0.33)
	U	0.79 (±0.06)	12.40 (±3.86)	2.47 (±0.49)	0.52 (±0.06)	3.05 (±0.68)	0.004 (±0.002)	2.65 (±0.51)	0.62 (±0.05)
% DIF R-U		−38.95	−29.59	−26.90	−10.50	−18.18	−9.52	−12.04	−26.57
PALENCIA	R	0.68 (±0.06)	12.11 (±4.48)	1.49 (±0.32)	0.33 (±0.03)	1.86 (±0.32)	0.003 (±0.001)	1.95 (±0.44)	0.54 (±0.06)
	U	0.85 (±0.07)	14.31 (±5.10)	1.77 (±0.33)	0.40 (±0.06)	2.53 (±0.37)	0.005 (±0.002)	2.09 (±0.42)	0.64 (±0.07)
% DIF R-U		−19.29	−15.39	−15.64	−16.93	−26.29	−36.29	−6.51	−16.09
SALAMANCA	R	0.48 (±0.05)	10.73 (±3.38)	2.47 (±0.46)	0.51 (±0.10)	2.39 (±0.44)	0.007 (±0.001)	2.40 (±0.56)	0.52 (±0.09)
	U	0.59 (±0.07)	9.73 (±2.92)	2.27 (±0.33)	0.61 (±0.09)	2.26 (±0.41)	0.006 (±0.002)	1.81 (±0.47)	0.59 (±0.06)
% DIF R-U		−17.94	10.25	9.14	−15.75	6.06	18.59	32.46	−11.76
SEGOVIA	R	0.77 (±0.05)	10.95 (±3.15)	1.90 (±0.16)	0.34 (±0.06)	1.57 (±0.26)	0.004 (±0.002)	1.94 (±0.45)	0.42 (±0.05)
	U	0.78 (±0.10)	8.90 (±2.49)	1.56 (±0.12)	0.28 (±0.06)	1.42 (±0.22)	0.002 (±0.002)	1.39 (±0.34)	0.36 (±0.03)
% DIF R-U		−1.03	23.04	21.69	21.92	10.97	94.96	39.81	15.53
SORIA	R	0.62 (±0.06)	13.24 (±4.07)	4.14 (±0.48)	0.40 (±0.08)	2.62 (±0.35)	0.018 (±0.004)	2.14 (±0.51)	0.72 (±0.03)
	U	0.69 (±0.05)	8.87 (±2.66)	3.14 (±0.53)	0.36 (±0.07)	1.88 (±0.30)	0.015 (±0.003)	1.54 (±0.44)	0.55 (±0.05)
% DIF R-U		−10.22	49.24	31.94	11.50	39.60	21.34	39.05	31.06
VALLADOLID	R	0.49 (±0.08)	7.06 (±2.26)	1.05 (±0.19)	0.26 (±0.04)	1.41 (±0.29)	0.003 (±0.001)	1.44 (±0.36)	0.36 (±0.05)
	U	1.02 (±0.08)	9.73 (±2.96)	1.59 (±0.29)	0.41 (±0.06)	2.21 (±0.49)	0.003 (±0.001)	1.87 (±0.49)	0.52 (±0.05)
% DIF R-U		−52.08	−27.40	−33.94	−37.76	−36.40	11.27	−23.28	−30.61
ZAMORA	R	0.70 (±0.07)	14.86 (±5.72)	3.31 (±0.43)	0.70 (±0.17)	2.67 (±0.49)	0.008 (±0.004)	3.24 (±0.94)	0.59 (±0.05)
	U	0.74 (±0.07)	10.81 (±3.92)	2.18 (±0.23)	0.47 (±0.07)	2.28 (±0.42)	0.010 (±0.007)	2.01 (±0.59)	0.48 (±0.02)
% DIF R-U		−5.81	37.46	51.68	47.10	16.85	−22.28	61.13	22.60
CASTILE & LEON	R	0.57 (±0.03)	10.41 (±3.40)	2.02 (±0.26)	0.43 (±0.05)	2.14 (±0.37)	0.006 (±0.001)	2.22 (±0.51)	0.49 (±0.04)
	U	0.81 (±0.05)	10.90 (±3.44)	2.04 (±0.28)	0.45 (±0.06)	2.41 (±0.40)	0.005 (±0.002)	1.98 (±0.48)	0.54 (±0.03)
% DIF R-U		−29.87	−4.45	−0.96	−4.23	−11.24	27.69	11.99	−9.99

**Table 4 antibiotics-14-01070-t004:** Average prescription rates of azithromycin through the PHC centers of the provinces of Castile and Leon and throughout the autonomous region over the period 2013 and 2023. % DIF R-U: percentile rural–urban difference. DDDs: Defined Daily Doses. HCCs: healthcare card-holders.

	DDDs Per 1000 HCCs Per Day RURAL	DDDs Per 1000 HCCs Per Day URBAN	% DIF R-U
ÁVILA	1.419 (±0.295)	1.194 (±0.280)	18.768
BURGOS	1.934 (±0.263)	1.961 (±0.251)	−1.371
LEÓN	1.942 (±0.425)	2.354 (±0.499)	−17.502
PALENCIA	1.149 (±0.239)	1.432 (±0.268)	−19.782
SALAMANCA	1.757 (±0.342)	1.508 (±0.294)	16.485
SEGOVIA	0.879 (±0.176)	0.714 (±0.141)	23.135
SORIA	1.801 (±0.275)	1.235 (±0.222)	45.823
VALLADOLID	0.909 (±0.181)	1.439 (±0.320)	−36.823
ZAMORA	1.830 (±0.334)	1.618 (±0.359)	13.040
CASTILE and LEON	1.513 (±0.264)	1.661 (±0.289)	−8.937

## Data Availability

The data on antibiotic prescriptions provided through Concylia were requested through its “Data Release Protocol”, which contains a “Confidentiality Commitment” and an undertaking to uphold Organic Law 3/2018, of 5 December, on the Protection of Personal Data and the guarantee of digital rights.

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
