# Peer review of "An Analysis of Primary Healthcare Antibiotic Prescription Rates Within Castile and Leon (Spain): 2013–2023"

_antibiotics, 2025, doi:10.3390/antibiotics14111070_

Round 1

Reviewer 1 Report

Comments and Suggestions for Authors

In this manuscript, the authors studied systemic antibiotic prescription rates in Basic Health Centres in Castile and Leon (Spain) during the period 2013-2023 and determined which sociodemographic variables might influence antibiotics prescribing. Despite the numerous limitations described by the authors in the article, this work is of relative interest, as it provides evidence to guide interventions and policies aimed at improving antibiotic use tailored to local realities, thus helping to curb the spread of bacterial resistance.

The introduction provides sufficient background and all references are appropriate and relevant to the research.

The materials and methods are well described. However, the inclusion of a geographic map would facilitate reading and understanding of the text, as well as data visualization and the location and orientation process.

The results are clear but not well-structured, as the discussion section should not contain figures or tables. These, along with the text describing the data shown, should be included in the results section. Therefore, the results and discussion need to be rewritten.

Additional comments:

In this reviewer's opinion, the word, although of Latin origin, should not be italicized.

Line 64: “Castilla y León” change to “Castile and Leon”.

Line 97: Please remove “Figure 1” from the title.

Line 361: In addition to nurses, could pharmacists also play a role?

Line 480: “a minimum was reached in 2023”. Please check.

Authors should write the full name of the microorganisms the first time they cite them.

Acronyms/Abbreviations/Initialisms should be defined the first time they appear in each of three sections: the abstract; the main text; the first figure or table. When defined for the first time, the acronym/abbreviation/initialism should be added in parentheses after the written-out form. After introducing the acronyms/abbreviations/initialisms, use the abbreviation by itself, without parentheses, throughout the rest of the section.

Author Response

I am providing a point-by-point response to the reviewer’s comments as a uploaded PDF file.

Reviewer 2 Report

Comments and Suggestions for Authors
  • The study addresses an important and timely issue—antibiotic prescription trends and their relation to sociodemographic factors, which is crucial given the global concern about antimicrobial resistance (AMR.
  • Suggestion for Improvement
  • The opening sentence of abstract is strong but could be more precise
  • Expand the background with global and national prescribing trends.
  • The statement “values in 2023 comparable to 2019” could be specified numerically for clarity.
  • Differences between provinces are noted, but exact variation ranges (highest vs. lowest prescribing provinces) would make the findings more impactful.
  • Refine language for clarity and conciseness.
  • Some sentences are long, with multiple clauses, reducing readability.
  • Phrases like “antibiotic prescription profile in primary healthcare” could be simplified.
  • Figures like “13.5% reduction” and “20.2% increase” are valuable but should be presented in a more consistent format, ideally with years and baseline values.
  • Some numbers (e.g., 28.06 DDD, 15.56–27.06 DDD) appear abruptly without context; a sentence of explanation would help.
  • References are cited but not always explained (e.g., “[5]” after high prescribing rates in Europe). The background would be stronger if each reference is linked to a clear finding.
  • “Fast turning into a global health emergency” sounds informal; could be rephrased to “is increasingly recognized as a global health emergency.”
  • The aim is at the end, which is fine, but the sentence could be shorter and sharper to clearly highlight what this study adds beyond previous national-level analyses.
  • Strengthen the justification for why Castilla y León is a relevant case study (aging index, rural distribution, intra-regional variability).
  • Explicitly highlight the gap this study addresses: While national studies exist, few have analyzed long-term (10 years) prescription trends at the regional and sociodemographic level.
  • Instead of listing each province separately, summarize with ranges and highlight notable outliers.
  • Always use DDD per 1000 HCC per day (avoid mixing DCD/DDD).
  • Shorten legends and move methodological details (definitions) to Methods.
  • Group findings into thematic patterns (seasonality, rural–urban differences, therapeutic subgroup differences, outlier provinces).
  • Shorten sentences, remove redundancy, and make transitions smoother.
  • Always refer to DDD per 1000 HCC/day.
  • Go beyond describing differences—discuss underlying mechanisms and implications.
  • Present in bullet-like structure for clarity.
  • Replace informal expressions with precise academic phrasing.
  • Add exact thresholds (e.g., <10,000 inhabitants = rural, as per prior study [6]).
  • State explicitly that exclusion of private healthcare may underestimate true antibiotic consumption.
  • Briefly describe trend analysis (e.g., linear regression, ANOVA, or descriptive only).
  • Even if exempt, specify that “no individual-level data were used; aggregated data were obtained from official sources in compliance with [protocol].”
  • Replace “Covid-1” with COVID-19.
  • Shorten long sentences and avoid redundancy for clarity.
  • Provide population thresholds or administrative definitions.
  • Clarify if analysis was purely descriptive. If no inferential statistics, state: “Only descriptive statistics (mean, SD) were calculated; no inferential analyses were performed.”
  • Replace “num.” with “number of,” and streamline software description.
  • Add a closing statement such as: “No individual-level data were used; all data were anonymized and aggregate.
  • Correct or rephrase the 2023 finding for clarity.
  • Explicitly state what these results mean for antibiotic stewardship and public health.
  • End with 1–2 practical recommendations or calls for future research.
  • Replace informal/uncertain terms with precise academic phrasing.

Author Response

(The authors gave the same response as above.)

Reviewer 3 Report

Comments and Suggestions for Authors

Major Comments

  1. The study is descriptive and observational, but some interpretations suggest causality (e.g., attributing decreases solely to PRAN or COVID-19). These should be phrased more cautiously or supported with statistical testing.

  2. Results are mostly descriptive. Inferential statistics (e.g., regression, time-series analysis, or multivariate models) would strengthen conclusions, especially about sociodemographic influences (aging, rurality).

  3. The study uses DDD per 1000 health-care card-holders per day (DCD) instead of DDD per 1000 inhabitants per day. While justified, the comparability with national PRAN data is only assumed. A sensitivity analysis or more discussion of potential biases would be valuable.

  4. Semi-urban areas were merged into “urban.” This may dilute true differences. Authors should justify this decision more clearly or acknowledge potential bias.

  5. The study notes that prescriptions cannot be linked to diagnoses. This is a major limitation and should be highlighted earlier in the abstract and introduction to avoid overinterpretation.

Minor Comments

  •  Some sentences are long and complex; simplifying for clarity would improve accessibility.

  • Some figures (e.g., Figure 3, multiple provincial time-series) are crowded; consider summarizing key comparisons.

  •  “Prescription rates” vs. “consumption” is sometimes used interchangeably; clarify consistently.

  •  Update citations where newer AMR/antibiotic consumption data are available (post-2023 if possible).

  • State causal claims more cautiously especially fi authors do not plan to conduct statistical analysis beyond descriptive

  • Consider including statistical modeling to analyze associations between sociodemographic variables and antibiotic prescribing if not causal inferential statistics

Author Response

(The authors gave the same response as above.)

Round 2

Reviewer 1 Report

Comments and Suggestions for Authors

Thanks to the authors for their responses and for revising their paper to address my previous comments and suggestions. However, some minor points still need to be addressed:

Lines 51-52: In my opinion, the increase in generics does not necessarily mean a decrease in antibiotic consumption.

Line 64: “Castile y Leon” change to “Castile and Leon”.

Line 303: Figure 9 should be moved to the result section.

Lines 326, 327 and 342: Please remove the period after the generic name of the species.

Please check that the citation of tables and figures is correct: line 188 (Table 3?) and line 204 (Table 1?). Please cite Figure 3 in the text.

Author Response

We sincerely thank you for your thorough second review of the manuscript. We hope that, once the minor errors you pointed out have been corrected, the work can be published in this great journal. Please find the detailed responses in the attached pdf file. The corresponding revisions and corrections highlighted in yellow and with track-changes in the resubmitted files. Please do not hesitate to contact us if any further clarifications or adjustments are needed; we remain fully available to address additional comments and ensure the manuscript meets the highest standards

Reviewer 3 Report

Comments and Suggestions for Authors

Dear Authors, 

Thank you for extensively revising the work, and you have put in a great amount of effort in doing so, good work. Thank you for considering the comments and am glad it was helpful. I have no further comments. 

Good luck!

Author Response

Dear Reviewer,
We sincerely appreciate your kind words and the time you dedicated to reviewing our work. Your constructive feedback, especially during the first round, was instrumental in improving the manuscript, and we truly value that you recognized the effort we put into addressing your comments. Your insights significantly strengthened the quality and clarity of the article. Thank you once again for your thoughtful contributions and support throughout this process.